



# Bacteria as paleoenvironmental proxies: the study of a cave Pleistocene profile

Catalina Haidău [1§*], Ionuț Cornel Mirea [1§], Silviu Constantin [1,2], Oana Teodora Moldovan [2,3*]

[1]Emil Racovita Institute of Speleology, Bucharest, 050711, Romania
[2]Centro Nacional de Investigación sobre la Evolución Humana, Burgos, 09002, Spain
[3]Emil Racovita Institute of Speleology, Cluj-Napoca Department, Cluj-Napoca, 400006, Romania

§These authors had an equal contribution.

*Correspondence to*: Catalina Haidău (haidau.catalina@gmail.com) and Oana T. Moldovan (oanamol35@gmail.com)

**Abstract.** Caves are well-known archives that preserve valuable information about the past, relevant for reconstructing past climates and environments. We sampled sediments from a 480 cm deep profile. 16S rRNA gene-based metabarcoding analyses were undertaken that complemented lithological logging, sedimentology, and Optically Stimulated Luminescence (OSL) dating. These analyses revealed different sedimentation conditions along the profile with various water inputs. The OSL age of the sediments places the profile between $74.7 \pm 12.3$ to $56 \pm 8$ ka (base to top). However, more recent reworking processes
(during LGM – Last Glacial Maximum paleofloods) in the upper and lower passages of the cave might have occurred. Bacterial compositions changed with depth, from soil bacteria (present in the upper part of the sediment profile) to thermophilic/sulfurous bacteria (abundant in the deeper samples of the profile). Considering the thermophilic bacteria, we could only assume their origin from a surface of hot sulfurous, old thermal springs, or sapropel sediments.

## 1 Introduction

Caves are known archives that preserve valuable information about climate in deposits and are relevant for the reconstruction of past climate and past environments (e.g., White, 2007; Nejman et al., 2018; Constantin et al., 2021; Bernal et al., 2023; Campaña et al., 2023). Caves are also known as systems with no primary production due to the lack of plants, low nutrient input from the surface, and low in situ production (Howarth and Moldovan, 2018; Kosznik-Kwaśnicka et al., 2022). That also means that the number of biological proxies (fossil bones, fossil invertebrates, pollen) in caves to be used in
paleoenvironmental and paleoclimatic studies is relatively low, and they might even be absent (Moldovan et al., 2011, 2016). Therefore, studying bacteria in cave deposits can open an avenue for paleoenvironmental research and fill some gaps related to presumed past processes in the absence of reliable proxies.

Bacteria are crucial in oligotrophic environments such as caves, as they can be primary producers and participate actively in biogeochemical cycles (Talà et al., 2021; Zada et al., 2021; Lange-Enyedi et al., 2022). Microbes regulate essential ecosystem



processes such as the biogeochemical cycling of micro- and macronutrients (Uroz et al., 2009; Pester et al., 2012) or are implicated in the formation (Yarwood et al., 2018; Domeignoz-Horta et al., 2021) or decomposition of organic matter (Krishna and Mohan, 2017; Prescott and Vesterdal, 2021). They can adapt and survive depending on abiotic and biotic factors (litter inputs, moisture, temperature; Castro et al., 2010; Wani et al., 2022).

Bacterial community and structure can change over time due to modifications in the physicochemical components of an
ecosystem, leading to the development of biogeographical patterns (Malard et al., 2019; Thomas et al., 2019; Bay et al., 2020; Ji et al., 2020). These patterns serve as indicators of past environmental changes. For example, the structure of bacterial communities present at deposition becomes preserved in sediment layers formed under changing environmental conditions (Frindte et al., 2020; Semenov et al., 2020; Barbato et al., 2022). Thus, such environmental conditions could be traced by reconstructing bacterial community structures from different sediment layers (Thomas et al., 2019; Frindte et al., 2020;
Semenov et al., 2020; Barbato et al., 2022).

Microbial paleoenvironmental studies on soil and lake/sea sediments are more common than in caves, highlighting the importance of their diversity as an indicator of ecosystem function and environmental conditions for reconstructing the past. For instance, Xu et al. (2022) found microbial communities in lacustrine sediments that provided valuable insights into past environmental and climate changes. The distinct vertical trends in microbial community structures, influenced by abrupt
environmental shifts, suggest that these communities responded dynamically to climatic events, such as aridification and cooling, around 8 million years ago. These shifts are also consistent with previous pollen evidence, indicating a transition from forest to steppe vegetation correlated with a significant uplift of the Tibetan Plateau. More et al. (2019) examined the microbial communities in the Black Sea sediments in response to substantial paleoenvironmental changes, mainly focusing on the transition around 5.2 ka. This study highlights bacterial composition changes driven by increased salinity. The research also
underlines key microbial metabolic processes, shift from methane metabolism before 5.2 ka to enhanced nitrogen and sulfur metabolisms. These changes correspond with the establishment of modern conditions in the Black Sea.

A study on paleosols (Frindte et al., 2020) that analyzed environmental DNA from different horizons within an arid paleosequence revealed key changes in microbial communities over time. The findings indicate bacterial abundance, diversity, and community composition decline with increasing soil depth and age. However, deviations from this trend were observed in
paleosol horizons with higher microbial diversity and abundance, suggesting that advanced soil formation processes may have preserved more diverse microbial communities. The study also identified specific microbial taxa associated with certain soil horizons, proving that some microbial communities from ancient environments remain detectable despite burial.

Regarding the caves, most studies on paleoenvironment focused on proxies such as stable isotopes (Waltgenbach et al., 2021; Weber et al., 2021), fossil bones (Berto et al., 2021; Mirea et al., 2021; Cruz et al., 2023), fossil invertebrates (Moldovan et
al., 2011, 2016; Buttler and Wilson, 2018; Romano et al., 2024; Osipova et al., 2022), or pollen (Prieto et al., 2021; Minckley et al., 2023).

Studies on cave microbes were performed regarding their diversity (Zhu et al., 2019; Dong et al., 2020; Dominguez-Moñino et al., 2021), associations (Dattagupta et al., 2009; Ma et al., 2021; Zhao et al., 2024), but little attention was given to their





potential as paleoclimate proxies (Epure et al., 2014, 2017; Yun et al., 2016). Furthermore, Epure et al. (2014, 2017) indicated

the potential of microbes from old cave sediment deposits in paleoenvironment and paleoclimate reconstruction. Zepeda Mendoza et al. (2016) explored the microbial communities within a speleothem, indicating their potential as past biodiversity archives. Metagenomic analysis on a speleothem in a cave near the sea found microbes related to soil and marine environments. Michail et al. (2021) revealed a complex and dynamic microbial community from a stalactite core composed of bacteria from the upper-ground environment. As indicated by specific bacteria, the evidence of seasonal climate variations emphasizes

environmental factors' role in shaping microbial composition over time. Overall, this research provided valuable insights into the microbial ecology of cave environments and highlights the need for further investigation into the role of microorganisms in cave deposits and paleoclimate reconstruction.

The scope of our study was to investigate the bacterial diversity from a 480 cm deep profile in Muierilor Cave, Romania, where no other biological proxy was found. This cave was studied for its evolution during the last 120 kyr. The combined

OSL, AMS$^{14}$C and sedimentology results, together with taphonomical analysis of the Pleistocene mammals' accumulation, indicated that most cave levels were already formed at ~120 ka with the lower levels functioning periodically as vadose cave passages where sediments from the Galbenul River were deposited (Mirea et al., 2021). The bacteria identified through the 16S rRNA gene-based metabarcoding were also compared to other proxies to help define past environments. Thermophiles and sulfur bacteria were amongst the high-abundance bacteria with depth, which raised questions about their occurrence since

the cave is characterized by a temperature much lower than their growth range. The possible sources of our samples are discussed, and the results strongly support the importance of investigating bacteria in old sediments, especially in the absence of other biological proxies. When cross-correlated with other proxies, our findings indicate the deposition conditions and water sources during the Pleistocene/Holocene, bringing new insights into the regional karst evolution.

## 2 Materials and methods

**2.1 Site description and sampling**

The geological settings of the Polovragi-Cernădia area are a part of the Parâng Mountains complex, where the basement is a combination of metamorphic pre-Alpine formation and granitic bodies (Hann et al., 1986), while the sedimentary deposits are represented by a mix of Upper Paleozoic and Mesozoic limestones, conglomerates, and Cenozoic deposits (gravel, sand, and clay) (Fig 1a). The limestones in this region belong to the Oslea-Polovragi formation, made of white-grey and white limestones

that can reach a thickness of 150-250 m, covering a surface of approximately 2 km$^2$ (Bandrabur and Bandrabur, 2010; Mirea et al., 2021).

Muierilor Cave (45°11′31.78″N and 23°45′14.07″E) (Fig. 1) is located at ~ 645 m a.s.l. in Baia de Fier, south-western Romania, being one of the most-visited show caves in the country due to its archaeological, paleontological, and mineralogical features (Fig. 1b). It is developed in Upper Jurassic-Lower Cretaceous limestone on the right side of the Galbenul Gorge. The cave

system is carved on four distinct levels, with a total length of more than 8000 m, and its cave levels are extended on an elevation



range of ~80 m. The significant parts of the cave include the Scientific Reserve (Level 1) and the Touristic Passage (Level 2) (Mirea et al., 2021). PMP2 (Fig. 1d, and Fig. 2) is a test pit of 1.5/1.5 m and with a depth of 480 cm, located at the northern end of the Urșilor Passage near the entrance towards the Hades Passage (Mirea et al., 2021).

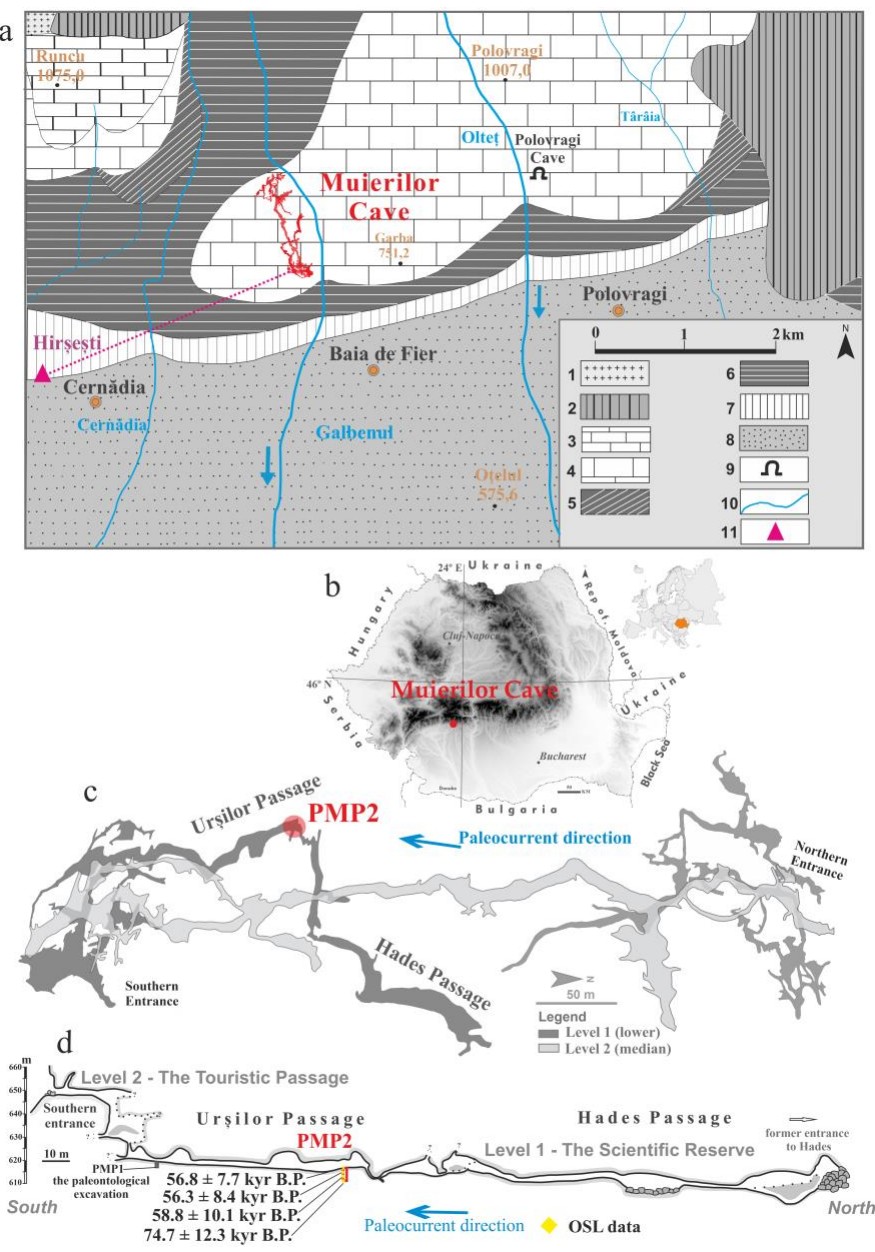

**Figure 1: The location of Muierilor Cave and details on the studied profile inside the cave. a. Geological map of the Polovragi - Cernădia area (modified after Diaconu et al., 2008): 1-Parâng Granites; 2-Metamorphic rocks; 3-Early Jurassic (limestone); 4-Late Jurassic (limestone); 5-Late Cretaceous (conglomerates, sandstones, and clays); 6-Early Miocene (marly clays); 7-Middle Miocene (sands and clays); 8-Late Miocene (gravels and sands); 9-Caves; 10-Rivers; 11-Location of the hot springs near Muierilor Cave (according to Ghenea et al., 1981); b. Location of the studied cave in Romania; c. A simplified map of the Muierilor Cave surveyed**



Preprint repository

**by the Emil Racovita Institute of Speleology and "Hades" Caving Club (Base map courtesy of Grigore Stelian); d. PMP2 profile with the OSL results (modified after Mirea et al., 2021).**

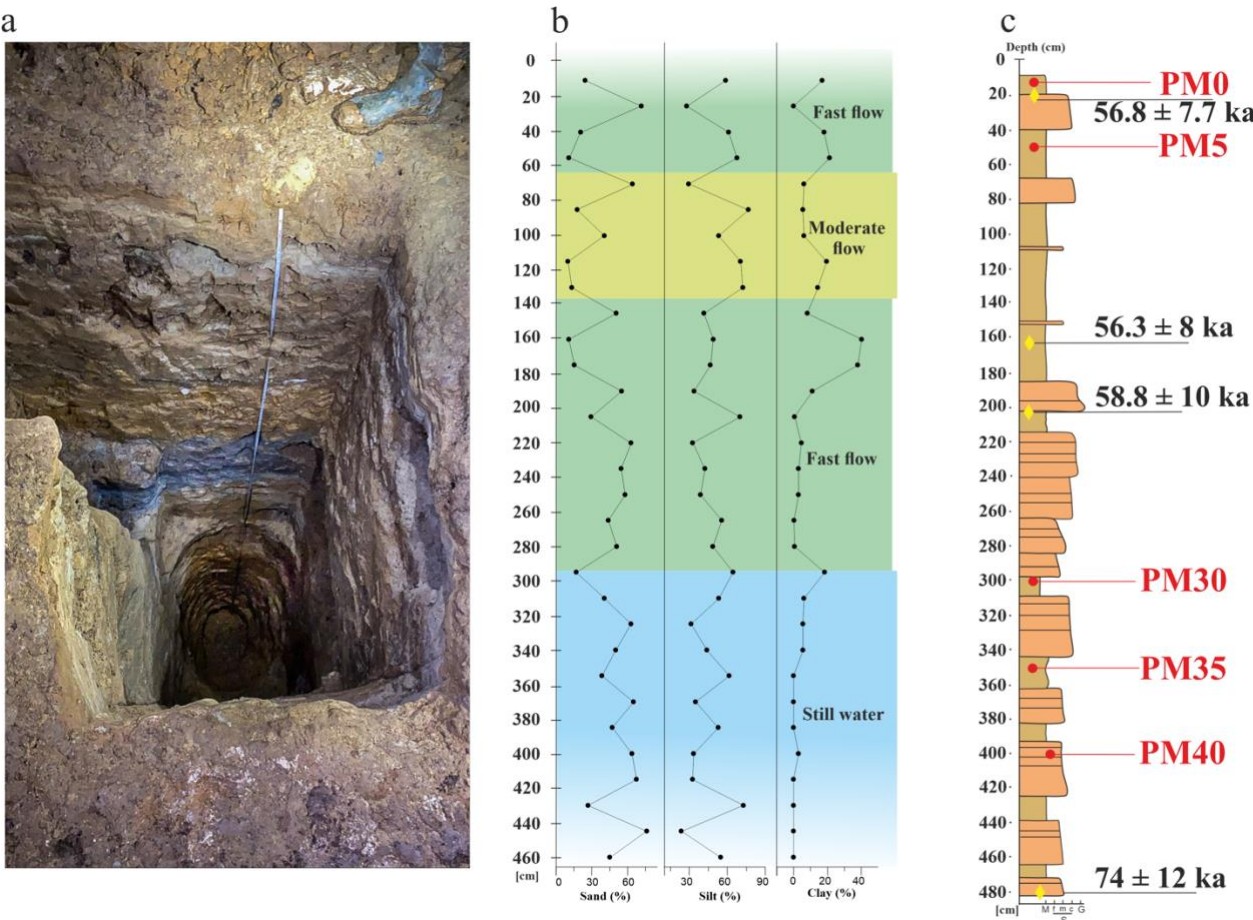

**Figure 2: The analyzed deposits in Muierilor Cave: a. Photo of the PMP2 section in Urșilor Passage seen from above; b. The grain size measurements (modified after Mirea, 2020 and Mirea et al., 2021); c. The position of the samples (red) and the OSL ages (yellow)**
**on the profile (modified after Mirea, 2020 and Mirea et al., 2021).**

The fossil record in Muierilor Cave is rich, and the long history of excavations (1950-2021) of the upper and lower levels of the cave revealed numerous species. The most significant fossil accumulation is in the Urșilor Passage because of primary and secondary thanatocoenosis (Mirea et al., 2021). The highest density of the fossil remains (~ 200 bones/m2) is reported near the PMP1 excavation site and decreases towards the PMP2 test pit (Fig. 1d.).

**2.2 Sediment analysis and chronological framework**

For the PMP2 profile, on-site lithological logging and analyses of sediments' grain size were undertaken by Mirea et al. (2021). Optically Stimulated Luminescence (OSL) was used to constrain the sediment input and the deposition events that occurred in the cave passages.



The PMP2 profile was excavated for sedimentology studies. It has a complex structure with sand, silts, and clay levels. It is located at the limit between the Urșilor and Hades passages (Fig. 1). Between 150 and 300 cm in depth; the sediments are alternating between sand and silt (Fig. 2), while higher amounts of clay appear in the upper sector, indicating a change in the source area. In contrast, in the topmost part, alternating clay and sands suggest the persistence of high-energy streams. The anisotropy of magnetic susceptibility (AMS) data (Fig. 2) showed that between 420 and 300 cm in depth, the sediments were deposited from still water (Tauxe et al., 1998), most likely due to a small lateral lake formed on the main cave stream. The following 150 cm (between 300 and 150 cm) are characterized by deposition under a high energy current flow, a moderate flow current deposited the segment between 150 to 50 cm, and the last 50 cm show a deposition in high, moderate currents (i.e. no particle entrainment). We, therefore, assume that the general flow direction was NE-SW with possible "apparent reversals", such as those due to vortex-type flows generated by cave wall topography.

The base of the sediments in the PMP2 profile has an OSL age of 74.7 ± 12.3 ka (Fig. 1d), while the upper 2 m of the section has a significantly younger OSL age of around 58 ka.

## 2.3 Sampling for DNA, extraction, and sequencing

For the microbiome analysis, sediment samples were taken directly into sterile Falcon tubes every 50 cm in the PMP2 profile. Sediment samples analyzed in this study were taken from the surface of the pit (PM0) and at -50 cm (PM5), -300 cm (PM30), -350 cm (PM35), and -400 cm (PM40) deep. Not enough genetic material for metagenomics could be extracted for the samples at -100 and -250 cm, most likely due to the high amount of clay. Clay is known for inhibiting microorganisms (McMahon et al., 2016). The samples were transported for further laboratory analysis in an icebox and kept in the freezer at –60°C until extraction. A quantity of 25 mg of sediment was used for DNA extraction.

We used FastPrep-24TM (MP Biomedicals) for cell disruption, and DNeasy PowerSoil (Qiagen) was used for genomic DNA extraction, according to the manufacturer's instructions. DNA was extracted in duplicates and was quantified using SpectraMax QuickDrop (Molecular Devices). Extracted DNA was used as a template and sent for MiSeq 16S V3-V4 Metagenome Sequencing using a commercial company (Macrogen Europe). PCR of the V3-V4 hypervariable regions of the bacterial and archaeal SSU rRNA gene was performed using bacteria-specific primers 341F (5'-CCTACGGGNGGCWGCAG-3') and 805R (5'-GACTACHVGGGTATCTAATCC-3'), according to Illumina's 16S amplicon-based metagenomics sequencing protocol.

## 2.4 Metabarcoding analysis and tests

Metabarcoding analysis was performed by a commercial company (Macrogen Europe) as follows: samples were analyzed using CD-HIT-OTU (Li et al., 2012) and rDnaTools (Schloss et al., 2009). Merging pairs of short reads was performed with FLASH (1.2.11) (Magoč and Salzberg, 2011). It is designed to merge pairs of reads when the original DNA fragments are shorter than twice the length of reads. CD-HIT-OTU is a multi-step pipeline to generate OTU clusters for ribosomal ribonucleic acid (rRNA) tags from 454 and Illumina platforms. CD-HIT-OTU and rDNATools were used to filter out short reads and





extra-long tails; filtered reads were clustered at 100% identity using CD-HIT-DUP. Chimeras were identified and removed. Remaining representative reads from non-chimeric clusters are clustered into OTUs at 97% OTU cutoff. Forward and reverse primers were removed, and for further analysis, reads with a minimum length of 250 nt and a maximum of 301 were retained. The sequencing depth varied between 79,534 and 126,869 sequences per sample, with a median of 112,595. The final dataset

consisted of a total of 2,692 OTU from 9 samples.

Taxonomic assignment and diversity statistics were performed by QIIME-UCLUST using NCBI targeted loci project databases 16S RefSeq version 20211127. Representative sequences from each OTU were used to assign taxonomy from phylum to species levels.

The raw data were deposited in the NCBI SRA Sequence Read Archive under the BioProject: PRJNA1161469, with

BioSample accessions: SAMN43780924, SAMN43780925, SAMN43780926, SAMN43780927, SAMN43780928, SAMN43780929, SAMN43780930, SAMN43780931, SAMN43780932.

Alpha diversity indices such as Shannon, Chao1, and Simpson's were used to express information about the composition of samples. Shannon considers the weight of each species in an ecosystem and gives a better description of its diversity (Konopiński, 2020). Simpson's diversity index estimates the probability that two randomly selected individuals will be

identical in a sample. The less diversity, the greater the likelihood that two randomly chosen individuals will be the same species (Simpson 1949; Zhou et al. 2020). Moreover, an abundance-based estimator of species richness, Chao1 index, was also calculated (Kim et al. 2017).

## 3 Results

### 3.1 Bacteria composition in the sedimentary profile

Only the domain *Bacteria* was kept for further analysis of microbial composition because the abundance of *Archaea* was very low (under 0.2%), with only one species (the ammonia-oxidizing *Nitrosopumilus ureiphilus*) present in PM0, PM30, PM35, and PM40. Except for PM40, the other samples provided enough material for duplicates (PM0, PM5, PM30, PM35) for which the mean abundances were used for further analysis.

From 2692 Bacteria OTUs, those with an abundance of over 1% were used for further analysis (see also Table A1). A total of

10 major bacterial phyla were identified in our samples (Fig. 3a), with *Proteobacteria* (22-62%) being the most abundant in all samples, followed by *Firmicutes* (PM0-20%; PM35-23%; PM40-30%), and *Actinobacteria* (PM0-10%; PM5-13%; PM30-20%; PM40-17%). The relative abundance of *Proteobacteria* decreased with depth, while *Firmicutes* and *Actinobacteria* relative abundances increased. *Cyanobacteria* appeared in surprising relative abundance in PM35 (9%).





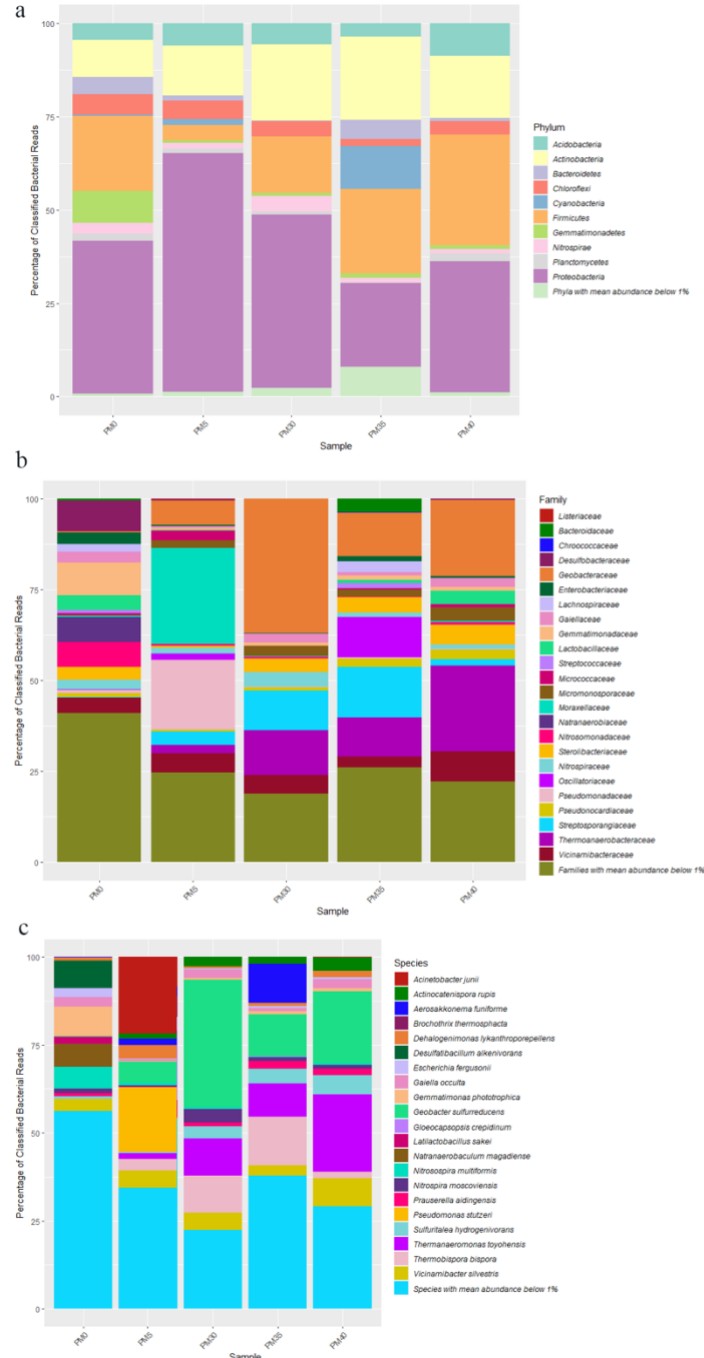

**Figure 3 The relative abundance of phyla (a), families (b), and species (c) in the analyzed sediment samples of Muierilor Cave (abundance >1%).**

The family composition (Fig. 3b) differed when comparing the upper with the deeper samples. In PM0, *Gemmatimonadaceae* and *Desulfobacteraceae* (8%), and *Natranaerobiaceae* (6%) were the most abundant, while in PM5 *Moraxellaceae* (26%),



*Pseudomonadaceae* (19%), and *Geobacteraceae* (6%). With deeper samples, the family relative abundances were quite similar, with *Geobacteraceae* (PM30-36%; PM35-12%; PM40-20%), *Thermoanaerobacteraceae* (PM30-12%; PM35-11%; PM40-23%) and *Sterolibacteriaceae* (PM30-3%; PM35-5%; PM40-5%) amongst the highest. *Streptosporangiaceae* (PM30-11%; PM35-14%; PM40-2%) and *Vicinamibacteraceae* (PM30-5%; PM35-3%; PM40-8%) were present in the highest abundance in the two samples. *Oscillatoriaceae* was the third highest in PM35 (11%; very low in the other samples), while

*Sterolibacteriaceae* in PM40 (5%; PM30- 4%; PM35-5%).

The most abundant species (Fig. 3c) from surface sample PM0 were *Gemmatimonas phototrophica* (7%), *Desulfatibacillum alkenivorans* (7%), and *Natranaerobaculum magadiense* (6%), while in PM5 (-50 cm) *Acinetobacter junii* (21%), *Pseudomonas stutzeri* (18%), and *Geobacter sulfurreducens* (6%) where the most abundant. As in the case of family abundances, species relative abundances are similar in deeper samples, with *Geobacter sulfurreducens* (PM30-28%; PM35-

10%; PM40-18%) and *Thermanaeromonas toyohensis* (PM30, PM35-8%; PM40-19%) present in all three samples with high abundances. *Thermobispora bispora* (PM30-8%; PM35-12%) was in high abundance in the first two bottom samples (-300 cm; -350 cm) while *Sulfuritalea hydrogenivorans* in the last two bottom samples (PM35-4%; PM40- 5%; -350 cm; -400 cm). Also, *Nitrospira moscoviensis* was found in high abundance in PM30 (3%; PM35, PM40-0.1%), and *Actinocatenispora rupis* in PM40 (3%; PM30, PM35-2%). *Vicinamibacter silvestris* was one of the most abundant bacteria throughout our profile (3-

7%). A cyanobacterium *Aerosakkonema funiforme* in high abundance in PM35 (9%; PM0-0.1%; PM5-1%; PM30-0.04%) was absent in PM40.

The Chao1 (Table 1) indicated that species richness was relatively higher in the surface samples (PM0 and PM5) than in deeper samples (PM30, PM35, PM40). The Shannon and Simpson diversity indices showed that surface samples (PM0) had a more diverse bacterial community than deeper samples (PM5; PM30, PM35, PM40).


**Table 1 Diversity indices of sediment samples from Muierilor Cave.**

| Sample | Diversity indices | | |
|---|---|---|---|
| | Chao1 | Shannon | Simpson |
| PM0 | 342.49 | 4.57 | 0.97 |
| PM5 | 395.15 | 3.26 | 0.82 |
| PM30 | 266.04 | 3.60 | 0.92 |
| PM35 | 231.71 | 3.66 | 0.93 |
| PM40 | 211.5 | 3.73 | 0.92 |



## 3.2 The distribution of bacteria in the sedimentary profile with depth/age

When investigating the bacterial distribution with depth (Figs. 3, 4, and 5) in the sedimentary profile, we discovered abundant species involved in biogeochemical cycles (S, Fe) or specific to different environments (soil, water, human), even extreme ones (thermophilic, halophilic).


**Figure 4 Distribution of bacteria categories with depth in the PMP2 sediment profile of Muierilor Cave.**



The family and species abundance in the profile first two depths (PM0; PM5) comprised bacteria commonly found in surface environments. Since some of them were identified chaotically throughout the profile, a plausible explanation could be the reworking of the sediment. For example, *Roseisolibacter agri*, an agricultural soil bacterium (Pascual et al., 2018), was found only in PM5 and PM35. Moreover, the presence of halophilic and halotolerant bacteria (*Aidingibacillus halophilus* - PM30, PM40; *Algiphilus aromaticivorans* – PM5, PM30; *Halomonas lactosivorans* and *Saccharopolyspora deserti* – in all samples) could be linked to the rhizosphere, because it may have plant growth promoting characteristics (Reang et al., 2022). Furthermore, another link to the surface environments might be the presence of animal or human-related bacteria. Such bacteria were found in higher abundance in PM0 and very low or absent in the other samples. For instance, *Escherichia fergusonii*, which causes diseases in humans and animals (Gaastra et al., 2014), was found in all samples, with the highest abundance in PM0 (2%; and under 0.4% in the other samples); human gut bacteria (Pianta et al., 2017; Hosomi et al., 2022) such as *Blautia wexlerae* and *Prevotella copri* were found in low abundance in PM0 and PM35, while *Megamonas funiformis* previously found in human faeces (Sakon et al., 2008), was found in PM0, PM5, and PM35.





**Figure 5 The distribution of bacteria at different depths in the analyzed Muierilor Cave deposits (left) compared to the isotopic oxygen and temperature variations (right; modified after North Greenland Ice Core Project, 2004); only the first 10 most abundant species in each sample (red-thermophiles, brown-involved in the S cycle) are represented.**

## 4 Discussion

The identified bacteria species from the first two sediment samples (PM0 and PM5 at 0 cm and -50 cm, respectively) are common in soils, indicating a direct input from the surface during the last fast flow episode(s). *Gemmatimonadaceae,*



*Desulfobacteraceae,* and *Natranaerobiaceae* were the most abundant families in PM0 and were commonly found in water, marine sediments, and soil (Chee-Sanford et al., 2019). Representatives of *Gemmatimonadaceae*, involved in the N cycle, are soil species (*Gemmatimonas aurantiaca*; Chee-Sanford et al., 2019) commonly found in agricultural soils (*G. kalamazoonesis*;
Jia et al., 2022) or are phototrophic (*G. phototrophica*; Koblížek et al., 2020). *Desulfobacteraceae* representatives were isolated from oil-polluted sediments, being involved in the S cycle (*Desulfatibacillum alkenivorans, Desulfatiferula berrensis*; Hakil et al., 2014; Ding et al., 2024), and *Natranaerobaculum magadiense* from *Natranaerobiaceae* was isolated from soda lake sediments, and it is obligately alkaliphilic, anaerobic, thermo-, and halotolerant (Zavarzina et al., 2013).

In PM5, representatives of one of the most abundant families (*Moraxellaceae*) such as *Acinetobacter tjernbergiae* was
commonly isolated from activated sludges (Yang, 2014), being involved in the P cycle and may have potential applications in the biological removal of phosphates (Täuber et al., 2022). *Psychrobacter aquimaris* is a halophilic bacterium isolated from seawater (Zhang et al., 2021). Denitrifying bacteria of *Pseudomonadaceae* were found in our samples (*Pseudomonas stutzeri*; Feng et al., 2020), and others previously isolated from marine sediments presenting an antagonistic activity (nitrification; *P. glareae*; Romanenko et al., 2015). *Geobacteraceae* representatives were mostly isolated from sediments except for
*Geoalkalibacter subterraneus*, which was isolated from petroleum reservoir water (Greene et al., 2009). Representatives of this family are metal-reducers such as *G. subterraneus* - an anaerobic Fe(III)- and Mn(IV)-reducing bacterium (Greene et al., 2009); *G. ferrihydriticus* - an alkaliphilic, iron-reducing bacterium, isolated from lake sediments (Zavarzina et al. 2020); *Geobacter argillaceus* - a Fe(III)-reducing bacterium, isolated from subsurface kaolin strata (Shelobolina et al., 2007) and *G. sulfurreducens* is capable of reducing different forms of Fe(III), Mn(IV), U(VI), elemental sulfur, fumarate and malate (Engel
et al., 2020), and was isolated from surface sediment of a hydrocarbon-contaminated soil (Caccavo et al., 1994). *Vicinamibacter silvestris* (*Vicinamibacteraceae*), a soil bacterium isolated from subtropical savanna soil (Huber et al., 2016) and from agriculture and residential (park) soil (Kim et al., 2022), was found in high relative abundances in all samples.

With depth, there was also an increase in the relative abundance of *Firmicutes* and *Actinobacteria*. This might be due to their resistant spores (Hashmi et al., 2020; Hazarika and Thakur, 2020), enabling the persistence unaffected by the environment for
more extended periods (Hanson et al., 2012; de Rezende et al., 2013). For the deeper samples (PM30, PM35, PM40) that mark the transition from fast flow to still water, the relative abundances of families and species are quite similar amongst all three depths. *Thermoanaerobacteraceae* representatives are thermophilic, involved in different biogeochemical cycles, and were isolated from various substrates. Found in all three depth samples are *Brockia lithotrophica*, isolated from a terrestrial hot spring and is involved in the S cycle (Perevalova et al., 2013), *Thermanaeromonas toyohensis* isolated from a geothermal
aquifer (Mori et al., 2002), and *Moorella stamsii* previously isolated from a digester sludge (Alves et al., 2013). *Desulfovirgula thermocuniculi* (PM30) was previously isolated from a geothermal underground mine (Kaksonen et al., 2007), and *Carboxydothermus hydrogenoformans* (PM30; PM35) grows with CO as their sole carbon source, was previously identified from a hot swamp (Henstra et al., 2004; Wu et al., 2005). The abundance of this family increases with depth.

Along the profile (see also Fig. 5), there is a transition of abundant species from soil Fe-reducing species (0 and -50 cm) to
thermophilic involved in S cycle bacteria (-300 cm, -350 cm, and -400 cm), with the most abundant *Geobacter sulfurreducens,*



*Thermanaeromonas toyohensis,* and *Sulfuritalea hydrogenivorans*. The identified thermophilic bacteria involved in biogeochemical cycle of S cycle are *Thiobacter subterraneus* (PM0, PM5; found in a hot aquifer by Hirayama et al., 2005), *Thermodesulfovibrio hydrogeniphilus* (all samples; found in a hot spring by Haouari et al., 2008), *Thermosulfurimonas dismutans* (PM5, PM30; deep-sea hydrothermal vent; Slobodkin et al., 2012), *Thioprofundum lithotrophicum* (PM0, PM5,

PM30, PM40; hydrothermal field; Mori et al., 2011). Bacteria involved in both the Fe and S cycles are *Acidiferrobacter thiooxydans* (PM0; an acidophilic, thermo-tolerant, copper mine drainage; Ma et al., 2022), and in Fe cycle *Aciditerrimonas ferrireducens* (PM0, PM5, PM30, PM40; acidophilic, geothermally heated field related with fumaroles emitting sulfurous gasses; Itoh et al., 2011). Sulphur can naturally occur in caves, also due to the presence of fossil bones and organic decay (Onac et al., 2011; Audra et al., 2019; Misra et al., 2019; Haidău et al., 2022); its biogeochemical cycle being driven by various

microbial metabolic activities, including sulphate reduction and oxidation (Holmer and Storkholm, 2001; Takahashi et al., 2011; Fike et al., 2015; Zhu et al., 2021). With specific bacteria in our samples, we would conclude that their source must be a hot sulfurous environment around the cave.

*Streptosporangiaceae* with the thermophilic representative, *Thermobispora bispora*, was present in all samples but with higher abundance in PM30 and PM35. This bacterium was isolated from soil (Slobodkina et al., 2017). *Sterolibacteriaceae*

representative *Sulfuritalea hydrogenivorans*, previously isolated from freshwater lakes, increases in abundance with depth. It can oxidize thiosulfate, sulfur, or hydrogen and degrade aromatic compounds (Sperfeld et al., 2019). *Nitrospira moscoviensis* was previously isolated from a heating system and was reported to be moderately thermophilic (Edwards et al., 2013).

The cyanobacterium *Aerosakkonema funiforme* (Oscillatoriaceae), found in all samples except for PM40, was previously isolated from a mesotrophic water reservoir (Thu et al., 2012), and from a hot spring microbial mat (Moreno et al., 2023),

indicating its survival at high temperatures.

Additionally, lower abundance thermophiles were identified in the lower samples, like *Thermanaerothrix daxensis, Caldilinea tarbellica* found in deep hot aquifers (Grégoire et al., 2011), or in hot springs *Thermoanaerobaculum aquaticum* (Losey et al., 2013), *Thermincola carboxydiphila* (Sokolova et al., 2005), and *Carboxydothermus islandicus* (Novikov et al., 2011). Bacteria tolerating high temperatures, such as *Gaiella occulta* (deep mineral water aquifer; Albuquerque et al., 2011), were also found.

A hydrogeological map of the area (Ghenea et al., 1981) included several mineral springs near Muierilor Cave, including thermal springs (Săcelu and Ciocadia) at less than 20 km distance. Some of them are well-known in the region for having therapeutic properties (Săcelu). In contrast, others were hard to identify in the field because of their low flow rate (Hîrșești see Fig. 1a). Direct proof of the existence of a thermal spring near Muierilor Cave (upstream of Galbenul River) is difficult to demonstrate due to the complex morpho-dynamic evolution of the river slopes. By the abundance of thermophilic bacteria, we

presume that the hot spring was present in the area and was the source of old input(s) of water in the cave. Although high concentrations of S and Fe can originate from fossil bones and organic decay in caves (Audra et al., 2019; Misra et al., 2019; Haidău et al., 2022), the thermophiles point to a different possible source, a hot spring.





**Figure 6 Digital elevation model of the relief near Muierilor Cave (a) with the hypothetical position of the paleolake relative to**
**Muierilor Cave and the flow direction during periods of high-water input from the upstream mountains (b; modified after Lupu &**
**Ion, 1962).**






Nevertheless, we do not rule out other possible sources and inputs, such as lacustrine organic sediments near the cave system. In certain conditions, sapropel sediments may form in small freshwater lakes (Leonova et al., 2019). Lupu & Ion (1962) reported the presence of a former lake upstream of the cave system, with intermittent inflows in the cave passages related to

the water availability from the snow and ice melting in the high mountains (Parâng Mountains; Fig. 6). Sapropels are characterized mainly as biogenic lake sediments, sludge sediment composed of organic matter and traces of clay, sand, or calcium carbonate (Leonova et al., 2019) with high concentrations of S amongst others (Mg, Fe, Ca) (Taran et al., 2018; Bogush et al., 2022). Moreover, Bogush et al. (2022) found that sulphate-reducing bacteria in a sapropel core from a lake near Baikal increased with depth, probably because such bacteria are important decomposers of organic matter. Thermophilic

bacteria are crucial in decomposition, especially when temperatures reach 700C for several weeks (Finore et al., 2023). Furthermore, S in our samples could result from the intensive decomposition, thermophiles being active in organic matter mineralization, and releasing inorganic nutrients (González et al., 2023).

The extensive clay deposits in the cave passages can also be related to the inputs from the former lake upstream of the cave system, a possible low-flow episode(s). Even though the OSL uncertainties span thousands of years, other proxies (e.g., fossil

remains, speleothems) dated from the cave passages constrained sediment deposition with the flooding events from the MIS 5 through the Holocene (Mirea et al., 2021). These sedimentation stages (episodes) can be associated with different climate events from the MIS 5 to the Holocene (Pleistocene), with warmer periods characterized by water and sediment input in the cave.

Microbial communities in caves are shaped by the constant input from the surface environments (Wu et al., 2015). Therefore,

the possibility of such bacteria being sourced from the surface is high. Thermophiles are thought to survive only in high-temperature habitats like compost heaps (Finore et al., 2023), hot springs (Benammar et al., 2020; Kochetkova et al., 2022), or deep-sea hydrothermal vents (Miroshnichenko and Bonch-Osmolovskaya, 2006; Zeng et al., 2021). However, they were also found in cool and temperate soils (Portillo et al., 2012; González et al., 2015; Santana and González, 2015; Santana et al., 2020; Milojevic et al., 2022), strengthening the idea of microbial dispersal and the possibility of tracking their movement

(Müller et al., 2014; Rime et al., 2016; Comte et al., 2017; Bell et al., 2018; Walters et al., 2022). Thermophiles can disperse on short or long distances from hot sources by water or wind (Portillo and González, 2008; Hubert et al., 2009; Perfumo and Marchant, 2010; Portillo et al., 2012; Bell et al., 2018). Soil is also a possible source. Thermophiles in temperate soils were considered vegetative viable organisms (Portillo et al., 2012; González et al., 2023), with the potential involvement in biogeochemical reactions (González et al., 2015, 2023). Recent studies on soil microbiota have included thermophiles as a

permanent component despite their strict ecological requirements (Portillo et al., 2012; Santana and González, 2015; González et al., 2023). Thermophiles that inhabit the upper soil layers are believed to grow and show significant enzymatic activity during hot days (>300C) to produce and stock extracellular enzymes that can help their activity under stress conditions (such as lower-temperature, dryness) (Milojevic et al., 2022; Gomez et al., 2021). For example, thermophiles showed enzymatic activity for more than 100 days per year at around 370N in Seville, Spain, and even only 1-2 hot days per year at 520N in

Cambridge, UK (Santana and González, 2015). There are ~ 40 hot days in Romania per year (Micu et al., 2015), and



thermophilic bacteria can survive in the soil. During extreme events, the thermophilic enzymes could decompose soil organic matter into smaller compounds (Santana et al., 2020), releasing N as ammonium (Portillo, et al., 2012) and S as sulfate (Portillo et al., 2012; Santana et al., 2021), at a higher rate than soil mesophiles (Portillo et al., 2012), indicating that S cycle in soils is performed mainly by them (Santana et al., 2021). A possibly high abundance of thermophiles in the soil could explain their
high abundance in the cave.

The presence of bacteria involved in Fe and S cycles in all our samples that date from the last interstadial could have different explanations. The deposit of fossil bones, or guano (Misra et al., 2019; Haidău et al., 2022) in the cave can be an essential source of these bacteria. Detrital clay (Audra et al., 2019) can be another source of these bacteria. The depositional condition of the fossil remains from the Urșilor Passage contributed to a rapid burial (e.g., fast flow phases), resulting in a slow diagenetic
process with few mineral exchanges (mostly apatite-related minerals) on long-term sedimentation. Different minerals in the upper levels were related to phosphate-rich deposits (bone and guano degradation; Haidău et al., 2022) at the same level as the studied profile. The interconnected passages of the cave on different levels (upper and lower) by shafts can influence the concentration of various minerals by the mixed sediment inputs. Not only the reworking processes during sedimentation inside the cave (Mirea et al., 2021) can complicate the sedimentation structure of the cave deposits, but also the intermittent link
between the different levels of the cave with distinct depositional settings.

Moreover, the bacterial abundance growth with depth could be correlated with the age of the sediments and be linked to more stable phases of the cave passages evolution when the sedimentation processes developed under a slow energy environment (Mirea, 2020; Mirea et al., 2021). Mirea et al. (2021) showed that the top sediments within the Urșilor Passage are linked with the warm condition of the Bølling–Allerød interstadial, the last inflow around ~14.7 ka. Thus, it explains the different bacterial
compositions correlated with sediment type and age.

**5 Conclusions**

The bacterial composition of a 480 cm deep profile in Muierilor Cave presented a clear difference between the upper (PM0, PM5) and bottom (PM30, PM35, PM40) samples. The composition changes with depth, from the dominance of soil-specific, Fe-reducing bacterial species to the dominant thermophilic, involved in S cycle bacteria. The presence of bacteria involved in
Fe and S cycles can be due to the presence of an abundance of fossil bones in the cave, probably brought inside the cave together with the sediments during the episodic paleofloods events associated with the end of MIS 5a and MIS 3 (Pleistocene). Thermophiles found in higher abundance in the lower part of the profile could originate from a warm water source in the area or from the soils above the cave during a warmer period. Still, their origin is yet to be determined. The presence of lacustrine organic sediments (sapropelic sediments) near the cave system can also be considered.


The study shows that bacteria in cave deposits can be used in a multi-proxy archive to understand sediment sources and the climate during deposition, as was proposed for other cave sites and organisms (Epure et al. 2014, 2017; Moldovan et al. 2011,



2016). It shows that for old sediments with complex depositional histories, Bacteria can offer new information at the time of deposition that can support or add to the entire understanding of the paleoenvironments.

**Appendix A**

Table A1. The most abundant bacteria in the analyzed sediments (PM0, PM5, PM35, PM40).

| Phylum | Class | Family | Species | PM0 | PM5 | PM30 | PM35 | PM40 |
|--------|-------|--------|---------|-----|-----|------|------|------|
| Proteobacteria | Gammaproteobacteria | Acidiferrobacteraceae | Acidiferrobacter thiooxydans | 0,01159 | 0,00000 | 0,00000 | 0,00000 | 0,00000 |
| Proteobacteria | Gammaproteobacteria | Moraxellaceae | Acinetobacter junii | 0,00005 | 0,21360 | 0,00000 | 0,00000 | 0,00139 |
| Proteobacteria | Gammaproteobacteria | Moraxellaceae | Acinetobacter lwoffii | 0,00216 | 0,01864 | 0,00012 | 0,00000 | 0,00000 |
| Proteobacteria | Gammaproteobacteria | Moraxellaceae | Acinetobacter tjernbergiae | 0,00000 | 0,02203 | 0,00030 | 0,00000 | 0,00003 |
| Actinobacteria | Actinomycetia | Micromonosporaceae | Actinocatenispora rupis | 0,00000 | 0,01374 | 0,02123 | 0,01673 | 0,03213 |
| Actinobacteria | Acidimicrobiia | Iamiaceae | Actinomarinicola tropica | 0,00056 | 0,00279 | 0,01318 | 0,00600 | 0,00868 |
| Proteobacteria | Gammaproteobacteria | Aeromonadaceae | Aeromonas veronii | 0,01963 | 0,00000 | 0,00251 | 0,00272 | 0,00505 |
| Cyanobacteria | | Oscillatoriaceae | Aerosakkonema funiforme | 0,00158 | 0,01623 | 0,00004 | 0,09270 | 0,00000 |
| Proteobacteria | Gammaproteobacteria | Algiphilaceae | Algiphilus aromaticivorans | 0 | 6,7854E-06 | 0,01122959 | 0 | 0 |
| Actinobacteria | Actinomycetia | Micrococcaceae | Arthrobacter globiformis | 0,00425 | 0,01988 | 0,00031 | 0,00001 | 0,00436 |
| Proteobacteria | Betaproteobacteria | Zoogloeaceae | Azoarcus olearius | 0,01979 | 0,00249 | 0,00362 | 0,00222 | 0,00721 |
| Firmicutes | Clostridia | Thermoanaerobacteraceae | Brockia lithotrophica | 0 | 0,00029856 | 0,01169357 | 0,00490068 | 0,00871121 |
| Chloroflexi | Dehalococcoidia | | Dehalogenimonas alkenigignens | 0,02778 | 0,00169 | 0,00370 | 0,00042 | 0,00075 |
| Chloroflexi | Dehalococcoidia | | Dehalogenimonas lykanthroporepellens | 0,00676 | 0,03443 | 0,00419 | 0,00861 | 0,01661 |
| Proteobacteria | Deltaproteobacteria | Desulfobacteraceae | Desulfatibacillum alkenivorans | 0,06513 | 0,00075 | 0,00000 | 0,00000 | 0,00000 |
| Proteobacteria | Alphaproteobacteria | Rhodospirillaceae | Dongia mobilis | 0,01460 | 0,00146 | 0,00000 | 0,00000 | 0,00000 |



| | | | | | | | | |
|---|---|---|---|---|---|---|---|---|
| Proteobacteria | Gammaproteobacteria | Enterobacteriaceae | Escherichia fergusonii | 0,02270 | 0,00151 | 0,00154 | 0,00384 | 0,00457 |
| Firmicutes | Clostridia | Oscillospiraceae | Faecalibacterium prausnitzii | 0,01828 | 0,00000 | 0,00000 | 0,00000 | 0,00000 |
| Proteobacteria | Alphaproteobacteria | Hyphomicrobiaceae | Filomicrobium fusiforme | 0,00022 | 0,00106 | 0,00021 | 0,00024 | 0,01043 |
| Fusobacteria | Fusobacteriia | Fusobacteriaceae | Fusobacterium mortiferum | 0,00000 | 0,00000 | 0,00000 | 0,02088 | 0,00000 |
| Fusobacteria | Fusobacteriia | Fusobacteriaceae | Fusobacterium perfoetens | 0,00000 | 0,00000 | 0,00000 | 0,01996 | 0,00000 |
| Actinobacteria | Rubrobacteria | Gaiellaceae | Gaiella occulta | 0,02332 | 0,00465 | 0,01890 | 0,00877 | 0,02123 |
| Gemmatimonadetes | Gemmatimonadetes | Gemmatimonadaceae | Gemmatimonas phototrophica | 0,07167 | 0,00254 | 0,00425 | 0,00785 | 0,00835 |
| Proteobacteria | Deltaproteobacteria | Geobacteraceae | Geobacter sulfurreducens | 0,00194 | 0,05557 | 0,27898 | 0,10328 | 0,17567 |
| Firmicutes | Bacilli | Lactobacillaceae | Lactobacillus gallinarum | 0,00314 | 0,00007 | 0,00026 | 0,00023 | 0,03059 |
| Firmicutes | Bacilli | Lactobacillaceae | Latilactobacillus sakei | 0,01705 | 0,00027 | 0,00000 | 0,00000 | 0,00000 |
| Verrucomicrobia | Verrucomicrobiae | Verrucomicrobia subdivision 3 | Limisphaera ngatamarikiensis | 0,00000 | 0,00396 | 0,01008 | 0,00113 | 0,00020 |
| Bacteroidetes | Bacteroidia | Bacteroidaceae | Mediterranea massiliensis | 0,00000 | 0,00000 | 0,00000 | 0,01564 | 0,00000 |
| Firmicutes | Clostridia | Natranaerobiaceae | Natranaerobaculum magadiense | 0,05574 | 0,00000 | 0,00000 | 0,00000 | 0,00000 |
| Proteobacteria | Betaproteobacteria | Nitrosomonadaceae | Nitrosospira multiformis | 0,05252 | 0,00546 | 0,00361 | 0,00087 | 0,00554 |
| Nitrospirae | Nitrospira | Nitrospiraceae | Nitrospira japonica | 0,01163748 | 0,01065302 | 0,00085276 | 0 | 0,00077745 |
| Nitrospirae | Nitrospira | Nitrospiraceae | Nitrospira moscoviensis | 0,01187 | 0,00403 | 0,02985 | 0,00987 | 0,00895 |
| Bacteroidetes | Cytophagia | Fulvivirgaceae | Ohtaekwangia koreensis | 0,02221 | 0,00003 | 0,00010 | 0,00000 | 0,00000 |
| Firmicutes | Clostridia | Peptococcaceae | Pelotomaculum thermopropionicum | 0,00000 | 0,00001 | 0,01140 | 0,00062 | 0,00397 |
| Firmicutes | Clostridia | Peptostreptococcaceae | Peptacetobacter hiranonis | 0 | 0 | 0 | 0,01174147 | 0 |
| Actinobacteria | Actinomycetia | Pseudonocardiaceae | Prauserella aidingensis | 0,00750 | 0,00102 | 0,00738 | 0,01932 | 0,01634 |
| Proteobacteria | Alphaproteobacteria | Xanthobacteraceae | Pseudolabrys taiwanensis | 0,00241937 | 0,01039518 | 0 | 0 | 0 |



| Proteobacteria | Gammaproteobacteria | Pseudomonadaceae | Pseudomonas stutzeri | 0,00000 | 0,18129 | 0,00000 | 0,00000 | 0,00000 |
|---|---|---|---|---|---|---|---|---|
| Firmicutes | Bacilli | Sporolactobacillaceae | Scopulibacillus darangshiensis | 0,00501991 | 0,0008346 | 0,00384998 | 0,01048405 | 0,00664113 |
| Proteobacteria | Betaproteobacteria | Sterolibacteriaceae | Sulfurisoma sediminicola | 0,02209 | 0,00028 | 0,00109 | 0,00003 | 0,00000 |
| Proteobacteria | Betaproteobacteria | Sterolibacteriaceae | Sulfuritalea hydrogenivorans | 0,00524 | 0,00245 | 0,02696 | 0,03618 | 0,04703 |
| Proteobacteria | Deltaproteobacteria | Syntrophaceae | Syntrophus aciditrophicus | 0,01895 | 0,00030 | 0,00000 | 0,00000 | 0,00000 |
| Firmicutes | Clostridia | Thermoanaerobacteraceae | Thermanaeromonas toyohensis | 0,00000 | 0,01527 | 0,08138 | 0,08227 | 0,18953 |
| Chloroflexi | Anaerolineae | Anaerolineaceae | Thermanaerothrix daxensis | 0,00070 | 0,00008 | 0,01310 | 0,00637 | 0,00078 |
| Actinobacteria | Actinomycetia | Streptosporangiaceae | Thermobispora bispora | 0,00017 | 0,03119 | 0,08183 | 0,12040 | 0,01628 |
| Chloroflexi | Anaerolineae | Anaerolineaceae | Thermomarinilinea lacunifontana | 0,00344139 | 0,00278879 | 0,00671799 | 0,00170065 | 0,01077193 |
| Planctomycetes | Planctomycetia | Thermoguttaceae | Thermostilla marina | 0,01045231 | 0,00196097 | 0,00111602 | 0 | 0 |
| Acidobacteria | Vicinamibacteria | Vicinamibacteraceae | Vicinamibacter silvestris | 0,02997 | 0,04494 | 0,03774 | 0,02585 | 0,06935 |

**Data availability**

The raw data were deposited in the NCBI SRA Sequence Read Archive under the BioProject: PRJNA1161469, with
BioSample accessions: SAMN43780924, SAMN43780925, SAMN43780926, SAMN43780927, SAMN43780928,
SAMN43780929, SAMN43780930, SAMN43780931, SAMN43780932.

**Author contribution**

OTM designed the study; CH, OTM and ICM wrote the manuscript; CH made the extractions; ICM described the samples;
ICM and SC made the paleoclimatic interpretation; all authors read and approved the manuscript.

**Competing interests**

The authors declare that they have no conflict of interest.



**Acknowledgments**

We are grateful to Alexandra Hillebrand-Voiculescu and Luchiana Faur for helping us with the sampling campaign and suggestions. We are also thankful to Stelian Grigore, Cristinel Fofirică, Arthur Dăscălescu, and Marius Iliescu ("Hades" Caving Club, Romania), the discoverers of the Hades Passage for providing the base map of the cave.

**Financial support**

This research was financially supported by the Ministry of Research, Innovation and Digitization grant, CNCS/CCCDI – UEFISCDI, project no. 2/2019 (DARKFOOD), within PNCDI III, the EEA Financial Mechanism 2014–2021 under project contract no. 3/2019 (KARSTHIVES 2), and the grant PN-III-P1-1.1-PD-2021-0262 (PALEOTRACE)

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
