# Peer review of "Bacteria as paleoenvironmental proxies: the study of a cave Pleistocene profile"

_EGUsphere, 2024_

## Community Comment (CC2)

[revised manuscript text omitted]

---

## Author Response (AR1)

We hope our contribution to Biogeosciences will be published in *this revised form*. The revised manuscript marks all the changes in red.

Based on your suggestions, the following are some comments concerning the changes made to the manuscript.

Dear Reviewer #1

*Haidău and co-authors in their study demonstrate the potential of bacteria as paleoenvironmental proxies in caves, especially in caves where traditional biological indicators are absent. The shift in bacterial composition with depth, from soil-specific to thermophilic and sulfur-associated bacteria, suggests changes in deposition conditions and water sources through time. This research highlights the need to consider bacteria as a valuable tool in multi-proxy paleoenvironmental reconstructions.*

*During the review I encountered a few issues with the MS. Firstly, by reading the MS I was not convinced that the authors fully consider how contamination and natural processes that happen after burial (i.e. taphonomic processes) might have affected the bacterial DNA and might have provide them with mixed results, although the expertise and experience of the authors is undoubted, I would be more convinced if parts of the text are rewritten highlighting the above.*

**The PMP2 profile and that part of the cave do not contain bones or taphonomic context. In the Material section, we mentioned the bones, and in Mirea et al. (2021), such taphonomic studies are described. Still, all the work was done in PMP1 (see Fig. 1d), representing a completely different situation than the studied PMP2. A few details were added in the text that will hopefully better describe the position of PMP2.**

*The MS will benefit if authors try to explore how this "sediment reworking" could impact the bacteria found at different depths.*

**The sediment reworking processes in Muierilor Cave are examined by Mirea et al. (2021), especially for PMP1, where the chronology established secondary depositions. In PMP2, there is no evidence for reworking or re-deposition; it is only for the input of sediments during key events like the paleofloods that occurred during warmer and wetter conditions in the region. No other significant processes, such as bioturbation or collapses, were identified. We agree that the term was not used properly and was deleted or replaced where PMP2 sediments were discussed.**

*I also believe that it should be stated in the manuscript how they tried to prevent contamination when collecting samples. This methodology/measures of protection can be valuable for others trying to perform such sampling. Also, the absence of this information might raise doubts about the accuracy of the findings, since modern bacteria could easily contaminate older sediment layers.*

**We added a paragraph to explain how contamination was avoided.**

*Another issue, in my opinion, is that the authors mainly base their idea of a hot spring as the source of thermophilic bacteria on indirect evidence. They haven't convincingly ruled out other possibilities, for example bacteria traveling long distances through epikarst.*

**The Polovragi-Cernadia karst area (where Muierilor Cave is carved) spans approximately ~2.5 km², with the limestone bar's width ranging from 2 km to 0.8 km and its thickness between 150 m and 250 m. Given this isolation of the limestone within surrounding magmatic and metamorphic rocks, it is unlikely that bacteria were transported over long distances (no more than 2 km) via the epikarst network. Moreover, PMP2 is in an area of the cave with no proof of present or former percolation and no speleothems. It is near a former siphon between two different cave passages. We added a phrase in the Material section to make this clear.**

*The authors don't fully discuss the limitations of using bacteria to reconstruct ancient environments. While the study shows great potential, it's important to acknowledge the challenges of interpreting bacterial data. For example, bacteria can go dormant, and their DNA can persist long after they're dead. This could mislead researchers about what the environment was like. The manuscript also doesn't consider how other environmental factors, like pH levels, nutrients, and even other microbes, might have shaped the bacterial communities. The manuscript would be more informative if it explored the functions of the bacterial communities. The authors briefly mention the roles of some bacteria in natural cycles, but they don't thoroughly examine the metabolic capabilities of the communities and how these might reflect past conditions. The manuscript's analysis could be improved with more advanced statistical methods to confirm the differences between bacterial communities found at different depths.*

**The extraction and high-throughput methods we used can identify active and dormant taxa. Testing the metabolism of bacteria can be misleading on such old sediments and it was not the idea of this study. The idea of the project was to investigate the bacterial diversity of a dated profile and compare the data with other proxies. If the results had been confusing and comparing different proxies impossible, more questions could have been proposed. Nevertheless, we consider that the results were informative for some of the events in the past, and a new figure (Fig. 4) with a statistical clustering was added. We also agree that future studies can consider more analyses/ experiments such as the ones recommended and make our study more interesting.**

*Finally, although I do understand that sampling microbes in a cave sequence is extremely demanding, I think that 5 samples only for a sequence 480 cm long covering ~20 k yrs are not enough to extract secure findings. I would appreciate it if the authors added to the text a small explanatory part explaining why they selected only 5 samples.*

**We sampled every 50cm in depth, but not enough genetic material could be extracted for the samples at -100, -150, -200, and -250 cm, most likely due to the high amount of clay, or other inhibitors that we could not remove. Details are given in the text, and we often face the problem of inhibitors in old and new cave sediments.**

*I believe that if all the above issues highlighted here addressed adequately by the authors the MS should be considered for publication from the journal.*

Dear Reviewer #2

*This interesting and well-documented paper presents a novel use of bacterial diversity as a paleoenvironmental proxy. However, it requires significant revisions before publication to address nomenclatural updates, methodological clarifications, and deeper discussion of the results.*

*The authors should take into account the International Code of Nomenclature of Prokaryotes, and update taxonomic names. A recent paper by See Oren and Garrity: Int. J. Syst. Evol. Microbiol. 71, 005056 (2021), changed the names of 42 phyla. Thus, Proteobacteria is now Pseudomonadota, Actinobacteria is Actinomycetota, Bacteroidetes is Bacteroidota, etc. Use the updated nomenclature throughout the text, figures, and tables.*

**We changed the names throughout the manuscript. Thank you for pointing out this error.**

*The use of 16S rRNA gene sequencing with the Illumina MiSeq platform has inherent limitations in resolving taxonomy to the species level due to the short-read lengths that target conserved regions of the gene. These limitations can result in ambiguities or misassignments when identifying microorganisms at the species taxonomic level. The authors should explicitly acknowledge these constraints in the manuscript and provide references to studies that highlight the challenges associated with species-level identification using Illumina MiSeq.*

**We also acknowledge that this approach has limitations in resolving closely related species due to the short-read lengths and the conserved nature of the targeted regions. Challenges have been documented in the literature (Bailén et al., 2020; Gehrig et al., 2022; Satam et al., 2023). The text was added in the Methods section.**

**We added a phrase in the Material section acknowledging these constraints.**

*The manuscript lacks a detailed discussion regarding bacterial community transitions with sedimentological and isotopic data across the depth profile, which are critical for interpreting environmental changes and depositional histories in cave systems. The authors present data on grain size, clay content, and isotopic variations but do not explore how these might correlate with microbial community structure. How do bacterial taxa shifts correlate with sedimentological data? Discuss how differences in grain size and clay content might influence bacterial diversity and composition. Integrate isotopic data with bacterial findings, as oxygen isotope variations in sediments could correspond to climatic conditions that influenced microbial inputs or activity. This approach will enhance the interpretation of the data and demonstrate the potential of using microbial proxies alongside traditional sedimentological and isotopic tools for reconstructing paleoenvironmental conditions.*

**The detailed sedimentological analysis of the PMP2 profile was presented and discussed in Mirea et al., 2021. In this study, we correlated the different types of sediments with flow rates based on the granulometry, as seen in Fig. 2. The isotopes have not been explored. We**

are unsure which isotopes you are referring to in this study, but such an exploratory study can be envisaged in the future. However, we are unsure how they will correlate if sediments were transported and redeposited inside the cave.

*Expand on the land use above the cave (e.g., agriculture, forestry, or urban development) to support your statements about surface inputs affecting microbial communities. Include data or references on land use and potential anthropogenic influences.*

**We added a paragraph in the Material section describing the surface and explaining the lack of anthropogenic impacts.**

*Replace the term "microbes" with "microorganisms" consistently throughout the manuscript. The term "microorganisms" aligns better with scientific and environmental contexts, whereas "microbes" is often associated with clinical and biomedical research.*

**We made the correction in the text.**

*M&M, section 2.4: "The sequencing depth varied between 79,534 and 126,869 sequences per sample, with a median of 112,595. The final dataset consisted of a total of 2,692 OTU from 9 samples." Clarify why results are presented only for five samples (PM0, PM5, PM30, PM35, and PM40) when the sequencing depth covered nine samples.*

**All samples were taken in duplicates, at every -50cm. There was enough material for the sequencing to be successful for PM0, PM5, PM30, PM35, but not for PM40, where only one of the two samples worked. Explanations were added in the text at Results section.**

*I suggest discussing inconsistencies, such as the unexpectedly high cyanobacteria abundance in certain depths: "A cyanobacterium* Aerosakkonema funiforme *in high abundance in PM35 (9%; PM0-0.1%; PM5-1%; PM30-0.04%) was absent in PM40."*

**The inconsistencies of bacteria throughout the soil profile could be explained by the reworking of the sediment due to the input of sediments during key events like the paleofloods that occurred during warmer and wetter conditions in the region. A phrase was added in the text.**

*Correct temperature notation (700C and 300C): "Thermophilic bacteria are crucial in decomposition, especially when temperatures reach 70°C for several weeks" and "During hot days (>30°C)…".*

**We corrected the text.**

**Thank you for the comments and suggestions that improved the manuscript.**